# Long term absence of invasive breast cancer diagnosis in 2,402,672 pre and postmenopausal women: *A systematic review and meta-analysis*

Winnifred Cutler[1]*, James Kolter[2], Catherine Chambliss[3], Heather O'Neill[4], Hugo M. Montesinos-Yufa[5]

1 Department of Research, Athena Institute for Women's Wellness, Chester Springs, Pennsylvania, United States of America, 2 Office of the Physician Director, Department of Obstetrics and Gynecology, Bryn Mawr Hospital, Bryn Mawr, Pennsylvania, United States of America, 3 Department of Psychology, Ursinus College, Collegeville, Pennsylvania, United States of America, 4 Department of Economics, Ursinus College, Collegeville, Pennsylvania, United States of America, 5 Department of Mathematics and Computer Science, Ursinus College, Collegeville, Pennsylvania, United States of America

he These authors contributed equally to this work.
* Cutler@Athenainstitute.com

**Data Availability Statement:** All relevant data are within the manuscript and its Supporting Information files.

# Abstract

## Background

Invasive Breast Cancer (IBC) risk estimates continue to be based on data collated from cancer registries, *i.e.*, retrospective research that excludes disease-free women. For women without a prior diagnosis, these estimates inflate both risk and screening frequency recommendations and inadvertently increase recently recognized harms from overdiagnosis and overtreatment.

## Objective

To estimate the likelihood that pre or postmenopausal women with no prior diagnosis will remain free of IBC in order to enable evidence-based screening recommendations.

## Methods

Prospective data from 21 studies of 2,402,672 women were analyzed, updating our previously published systematic search of 19 studies. This second systematic search included PubMed and The Cochrane Library from 2012 through April 2019. Inclusion criteria: only studies reporting the number of women enrolled, length of follow-up, and number of women diagnosed with IBC. Linear regression was used to estimate the percentage of women expected to remain free from an IBC diagnosis based on follow-up duration. To minimize non-response bias and selective outcome bias, only studies reporting outcomes for all enrolled women followed for similar, specific lengths of time were included. Sensitivity analyses confirm that the overall findings were unchanged by age at enrollment, menopausal status, screened women, variation in sample size, duration of follow-up, and heteroskedasticity.

**Funding:** The funder, Athena Institute, provided support in the form of salaries for author [WC] but did not have any additional role in the study design, data collection and analysis, decision to publish, or preparation of the manuscript. The specific roles of these authors are articulated in the 'author contributions' section.

**Competing interests:** Dr. Winnifred Cutler is the Founder, President, and sole owner of the Athena Institute for Women's Wellness. Coauthors Drs. Kolter and Chambliss also serve as members of Athena Institute's Research Advisory Group, without compensation, to evaluate research grants and protocols for clinical investigations from ethical, scientific and medical criteria. They have no role in Athena Institute's commercial activities. Athena is a biomedical research institute whose mission is centered on working to improve the quality of healthcare for women (http://www.athenainstitute.com/messages/whatisathena.html). This does not alter the authors' adherence to PLOS ONE policies on sharing data and materials.

## Results

The calculated percentage of women remaining IBC-free after follow-ups of 5, 10, 15, 20 and 25 years decreases uniformly by about one-fourth of one percent per year, *i.e.*, 0.255% (95% CI: -0.29, -0.22; p < .0001). At 25 years, the expected percentage of women with no invasive breast cancer is 93.41% (95% CI: 92.75, 94.07).

## Conclusions

Over 99.7% of pre/postmenopausal women with no prior diagnosis continued with no IBC each year, with 93.41% still free after 25 years. Our study supports the medical justification for reducing the frequency of mammograms for menopausal women with no prior IBC diagnosis.

## Introduction

Optimal healthcare decision-making depends upon patients and providers having accurate information about the potential benefits and risks of diagnostic and treatment procedures. Breast cancer screening pursuing early detection has harms as well as possible benefits [1–6] The harms justify careful selection of when and which candidates might best be excluded from such screening. This study examines prospectively the likelihood that pre and postmenopausal women not previously diagnosed with IBC will remain free of IBC for given durations.

In 2015, a multi-step systematic literature review through 2012, identifying and evaluating diagnostic outcomes of 2,305,427 pre and postmenopausal women with no prior diagnosis of breast cancer, was performed [1]. The percentage of women in each study that eventually was diagnosed with a first IBC was surprisingly low. Since that 2015 work was completed, the medical context has changed. The introduction of new medical technologies, such as 3D breast cancer screening, as well as the aging demographics of many developed countries, has increased the number of women at risk of overdiagnosis. Revised IBC screening guidelines have created greater ambiguity about best practices and a desire for more precise quantification of the screening needs of particular groups of women. Furthermore, the availability of new data from two long term studies now enables calculation of the likelihood of long-term absence of IBC diagnosis for pre and postmenopausal women with no prior history of breast cancer. Sensitivity analyses that correct for the inevitable heteroskedasticity permit estimation of the percentage of 2,402,672 women that are expected to remain free from a diagnosis of IBC over their next 25 years of life.

This study has achieved its aim of providing and interpreting accurate new data facilitating optimal decision-making for screening and treating invasive breast cancer. These findings should help to inform patients' and providers' decisions about the value of screening as well as avoiding it.

## Methods

### Literature search

Two multi-step, systematic search protocols exhaustively identified all qualified peer-reviewed published studies through 2012 and April 2019. See Online Supplements: S2 and S3 Tables,

and S1 Text which provide the specific details of each search encompassing 687,886 and 18,033 studies, respectively.

Every study met five inclusion criteria: 1) All the women were prospectively studied during their pre/postmenopausal years and had no invasive breast cancer history at enrollment; 2) The study specified the number of women enrolled; 3) The study specified the number of women diagnosed with a first invasive breast cancer as well as the number of women not diagnosed; 4) Study subjects were counted only once; and, 5) The duration of follow-up was clearly defined.

**The primary outcome of interest** for this review and subsequent data analysis was the total cumulative percentage of women still free of a diagnosis of invasive breast cancer via prospectively recorded data sets. To minimize potential sources of non-response and selective outcome bias, we included only studies where outcomes were reported for all enrolled women who were followed up for a similar, specific length of time.

The prospective method used in this study allows the estimation of long-term incidence-of, and absence-of, the first ever IBC diagnosis. In contrast, the CDC/SEER's retrospective methods using data from cancer registries are unable to estimate these crucial statistics because they lack the necessary information. Hospital registries collect data on cases of cancer, and they do not track women that move across medical systems, making the necessary raw data unavailable and therefore the calculation impossible using retrospective methods from cancer registry data. To correctly estimate the incidence of the first ever IBC diagnosis and unveil the absence of IBC, a prospective method is necessary.

Because of the way the data are collected and the difference in approaches, our prospective method and the retrospective registry method are not parallel to each other. Only IBC-free women enter our studies; only post-diagnosed women enter the CDC/SEER cancer registries.

To highlight the differences, Table 1 presents a side by side comparison of the two methods.

## Results

Twenty-one studies met all five inclusion criteria with each providing high-quality evidence for its duration of follow-up and its percentage of women who remained free of a diagnosis of IBC. We used regression analysis to model the relationship between the percentage of women who remained free from a diagnosis of IBC (the "survival rate") and the duration of study follow-up. Table 2 summarizes the design of 21 studies noting all interventions and separate outcomes when relevant to IBC diagnosis.

### Description of statistical analyses

The number of follow-up years and the sample size varied from one study to the next. The heterogeneity in the sample size across studies induces heteroskedasticity in the regression. In our sensitivity analysis, we solve the heteroskedasticity problem by complementing the ordinary least squares (OLS) regression analysis with a generalized least squares (GLS) analysis. Statistically, both OLS and GLS provide unbiased and consistent estimates of the relationship between the "survival rate" and the follow-up time; but GLS is more efficient [28]. These efficiency gains translate into higher precision, that is higher t-ratios, higher r-squares, lower p-values, and narrower confidence intervals.

The 21 studies covering 2,402,672 different women provided 24 separate outcomes. Studies 7, 8, and 19 each offered 2 qualifying outcomes because of each having a treatment and a control group. Each outcome states the percentage of the women in each study that remained free of a diagnosis of invasive breast cancer (IBC) at the end of that study's follow-up period. Table 3 provides the outcome data for each study.

**Table 1. Two ways of estimating the risk of invasive breast cancer diagnosis.**

| The Prospective Method | Retrospective Registry Method [1] |
|---|---|
| 1. The denominator is based on the precise number of eligible women enrolled (*does not use census estimates*) | 1. The denominator is based on the estimated number of eligible women in the census for the relevant age group and time period |
| 2. Based on the exact number of women diagnosed | 2. Based on the number of cases entered in cancer registry |
| 3. Women with cancer are counted once (*#case/#women represent incidence of 1$^{st}$ breast cancer*) | 3. Cases of cancer are counted, so one woman may account for several entries (*new tumours, new locations, different breast*) |
| 4. We know how long women were followed up (*accurate calculation of % diagnosed in each study and cumulative incidence for up to 25 years*) | 4. There is no follow-up as there were no prospectively recorded data. |
| Calculation of incidence is based on: # of women diagnosed with 1$^{st}$ IBC | Calculation of incidence is based on: # of cases of IBC |
| ------------------ | ------------------ |
| # of women enrolled in that study for a known follow-up period | # of women estimated from census data in a designated age-bracket and period |

**Table 2. The design of each of the 21 published studies enrolling ONLY women without a prior diagnosis of invasive breast cancer.**

| | Table 2 Trial | 21 Published Studies Enrolling ONLY Women without prior Diagnosis of Invasive Breast Cancer DESIGN DETAIL |
|---|---|---|
| 1 | UK Million Women [7] | Population-based **cohort study**: postmenopausal women, 25% with hysterectomy, and 11% with bilateral ovariectomy. All without prior cancer, recruited 1999–2001 to evaluate HRT impact in a breast cancer screening program with mammograms. 1.4% excluded because of prior cancer diagnosis. |
| 2 | Danish Nurses [8] | **Prospective cohort study** testing HRT relationship to breast cancer identified by linkage via unique ID# to Danish nationwide registries. Women with prior breast cancer excluded. Follow-up for incident cases started with the questionnaire in 1993, ended in 1999. |
| 3 | Melbourne [9] | Collaborative **cohort study**: history of HRT use among women postmenopausal at baseline within 5 years before enrollment. No prior invasive cancer (*in situ* lobular breast cancer permitted) in the 5 years before enrollment. Southern European migrants over-sampled to increase demographic range. |
| 4 | Finnish Registry I [10] | **Cohort study:** all Finnish women >50 using estrogens ≥6 months identified from national medical reimbursement register and followed for breast cancer using the national cancer registry. Women with prior breast cancer excluded. *(Since 1987, 90% of Finnish women have taken part in mass mammogram screening programs that offer a free-of-charge mammogram every 2nd year to all women between 50 and 60 with many also to age 65.)* |
| 5 | Finnish Registry II [11] | **Cohort study:** all Finnish women >50 using estrogen and progestogen ≥6 months identified from national medical reimbursement register and followed for breast cancer using national cancer registry. |
| 6 | French Cohort [12] | **Cohort study:** postmenopausal women, breast cancer in HRT users vs. non-users among women without prior breast cancer. |
| 7 | WHI [13] | **RCT** primary prevention trial, placebo vs. Prempro®: Women 50–79 with intact uterus and without prior breast cancer. Mammography screenings and clinical breast examinations at baseline and then annually. All IBC are counted, but number of patients is shown; Incidence = 1st incidence. |
| 8 | WHI II [14] | **RCT** primary prevention trial, placebo *vs.* Premarin®: Women 50–79 without uterus and without prior breast cancer. Mammography screenings and clinical breast examinations as WHI I. |
| 9 | The Gothenburg Breast Screening Trial [15] | **RCT,** screened vs. control groups, started in 1982 intending to show mammogram screenings reduced mortality in women without prior breast cancer. >49 yrs had 4; the rest had 5 invitations for screening every 18 months for 7 years. Follow-up data >5 years for breast cancer from cancer registries. Because the outcome did not differ at 14 years follow-up, the data are combined. |
| 10 | UK Trial of Early Detection of Breast Cancer (TEDBC) [16] | **Prospective cohort study** comparing screened women to other groups enrolled at the same time. Study does not indicate if those with prior breast cancer were excluded. Only the 2 cohorts of screened women are analyzed. |
| 11 | Australia Record Review [17] | **Retrospective cohort study** of referral center for testosterone supplementation for HRT users. Women with prior breast cancer excluded. All doses titrated individually. Baseline, then biannual mammograms. |
| 12 | Osteoporosis Fracture Study [18] | **Prospective cohort study** of community-based women age 70±5 with no prior breast cancer. Contrary to the hypothesis, baseline estradiol level did not predict subsequent breast cancer. |

(*Continued*)

**Table 2.** (Continued)

| | Table 2 Trial | 21 Published Studies Enrolling ONLY Women without prior Diagnosis of Invasive Breast Cancer DESIGN DETAIL |
|---|---|---|
| 13 | Italy ORDET [19] | **Prospective cohort study** of postmenopausal women with intact ovaries who contributed baseline blood samples to test whether baseline steroid levels were predictive of subsequent developing invasive breast cancer (histologically confirmed). None with prior cancer or liver disease. |
| 14 | N Y U [20] | **Prospective cohort study** of New York City postmenopausal women with no prior breast cancer who received screening for breast cancer at the time of blood sampling. |
| 15 | US Breast Cancer Detection Demonstration Project [21] | Retrospective record review of breast exams at 29 screening centers in 27 US cities providing total number of women without prior breast cancer, and number of women with breast cancer after screening, reported at the start of this **Prospective Cohort study** of physical activity and risk. 17% of the cases were *in situ* not shown in table. 89% Caucasian. |
| 16 | Sweden-Malmo [22] | 25- year **prospective cohort study** of women without prior breast cancer, all enrolled in 1976, followed until 2002, approx. 50% with *vs*. 50% without regular mammograms every 12 to 18 months during initial screening phase. Showing 19% overdiagnosis in women 55–69 with mammogram screening. Total cumulative incidence of breast cancer reported. |
| 17 | Norwegian Cohorts [23] | **Prospective cohort study:** Post randomization mammography screening (3 times) offered to all women <70. Showing 22% overdiagnosis in women 55–69. Total cumulative incidence of invasive cancer reported. |
| 18 | Swedish Two County Trial [24] | **Prospective cohort study:** randomized enrollment of women without prior breast cancer to active and passive screening in 1977. Designed to show long-term breast cancer mortality reduction from mammogram screening. Screening group results shown. 85% accepted screening for 7 consecutive years. 123 *in situ* cancers omitted. Author confirmed that only first cancer is recorded. |
| 19 | Canadian National Breast Screening Study [25] | **Prospective 25-year cohort study:** Women without prior breast cancer, randomized to 5 years of annual mammogram and clinical examination vs. clinical examination only by protocol-trained nurses and doctors. At 8 years, the number of invasive breast cancers found was equivalent. Mammograms produced an earlier lead-time, more biopsies and surgeries, but no mortality reduction. *(This is the only systematic screening study to compare protocol-trained annual clinical breast examination alone to clinical exam plus mammogram screening)*. Long-term follow-up data confirmed that annual mammography does not reduce mortality from breast cancer compared with usual, competent care. |
| 20 | Spain's PREDIMED Study [26] | **Prospective cohort study** of women 60 to 80 years at entry who were initially free from breast cancer but at higher risk for CVD disease to test dietary factors, glycemic load and IBC in postmenopausal women. |
| 21 | UK CTOCS: A Collaborative Trial of Ovarian Cancer Screening [27] | **Prospective cohort study** of only postmenopausal women, median age 64, residing in England with no prior breast cancer diagnosis addressed change in skirt size as a possible surrogate marker for weight change and risk of 1st incident breast cancer. They report that change was a better marker for risk than absolute skirt size. |

Fig 1 plots each of these 24 outcomes as one dot showing the duration and the percentage of women who remained free of a diagnosis of IBC at the conclusion of the study. Fig 1 illustrates that the percentages of women with no IBC diagnosis ranged from a high of over 99% for studies #1 (UK), #13 (Italy), and #20 (Spain) with follow up durations between 2.6 and 4.8

**Table 3. The results of each of the 21 published studies enrolling only women without a prior diagnosis.**

| Study | Table 3 Trial | Number of Women | Baseline Age | Years Studied | Number w/o Breast Cancer | Percent Cancer Free |
|---|---|---|---|---|---|---|
| 1 | UK Million Women Study | 1,084,110 | 50–64 | 2.6 | 1,074,746 | 99.14% |
| 2 | Danish Nurses Health | 10,874 | >44 | 6 | 10,630 | 97.76% |
| 3 | Melbourne Postmenopausal | 13,444 | 40–69 | 10 | 13,108 | 97.50% |
| 4 | Finnish Registry ERT | 110,980 | >50 | 8 | 108,809 | 98.04% |
| 5 | Finnish Registry E&P | 221,551 | >50 | 11 | 215,340 | 97.20% |
| 6 | French Cohort | 3175 | >50 | 13 | 3,070 | 96.69% |
| 7 | WHI I | 16,608 | 50–79 | 5.2 | 8,340 | 98.05% |
|  | Prempro 8506 |  |  |  | 7,978 | 98.47% |
|  | Placebo 8102 |  |  |  |  |  |
| 8 | WHI II | 10,739 | 50–79 | 7.1 | 5206 | 98.04% |
|  | Premarin 5310 |  |  |  | 5296 | 97.55& |
|  | Placebo 5429 |  |  |  |  |  |
| 9 | Sweden: The Gothenburg Breast Screening Trial | 51,611 | 39–59 | 14 | 50,102 | 97.08% |
| 10 | UK Trial of Early Detection of Breast Cancer | 39,773 | 45–64 | 7 | 39,314 | 98.85% |
| 11 | Australia Record Review of Postmenopausal Women | 508 | 35–84 | 5.8 | 501 | 98.62% |
| 12 | Osteoporosis Fracture Study | 9704 | >65 | 3.2 | 9,587 | 98.79% |
| 13 | Italy ORDET | 4040 | 40–69 | 3.5 | 4,015 | 99.38% |
| 14 | NYU Postmenopausal | 7063 | 35–65 | 5.5 | 6,942 | 98.29% |
| 15 | US Breast Cancer Demonstration Detection Program | 283,222 | 40–93 | 3.5 | 278,947 | 98.49% |
| 16 | Sweden-Malmo | 42,283 | 45–69 | 25 | 39,967 | 94.52% |
| 17 | Norwegian Cohorts | 229,256 | 50–64 | 6 | 225,259 | 98.26% |
| 18 | Swedish Two-Country Trial: Active Screened Group | 77,052 | 40–74 | 7 | 76,654 | 98.19% |
| 19 | Canadian National Breast Screening Study Groups Mammogram 44,925 Control group 44,910 | 89,835 | 40–59 | 21.9 | 41,675 | 92.77% |
|  |  |  |  |  | 41,777 | 93.02% |
| 20 | Spain's PREDIMED Study | 4010 | 60–80 | 4.8 | 3,978 | 99.20% |
| 21 | UK CTOCS | 92,834 | >50 | 3.19 | 91,744 | 98.83% |
|  | **Total Number of Women** | **2,402,672** |  |  |  |  |

years, to lows of 92.77% and 94.52% for the longest studies #19 (Canada), #16 (Sweden) with respective 21.9- and 25-years of follow up.

The linear regression statistical methods confirm what simply looking at the way the data array on the graph reveals: studies that lasted longer produced a linear (straight line) pattern of a slightly declining percentage of women still free of an IBC diagnosis. There is no obvious curving pattern that would occur if older pre and postmenopausal women were substantially more likely to receive the first diagnosis than younger women.

The estimated regression line indicates the predicted rate for women who remain free of an IBC diagnosis per year. The line's predicted slope of -0.267 indicates that the percentage of women expected to remain free from an IBC diagnosis decreased annually by 0.27 percentage points (p < .0001), *i.e.*, about one-fourth of one percent per year (95% CI: -0.31, -0.23). Consequently, in four years, the expected rate of absence of a diagnosis of IBC only declined by 1.1 percentage points. Overall, 93.25% of these 2,402,672 women would be expected to remain free of an IBC diagnosis after 25 years of follow-up (95% CI: 92.53, 93.97).

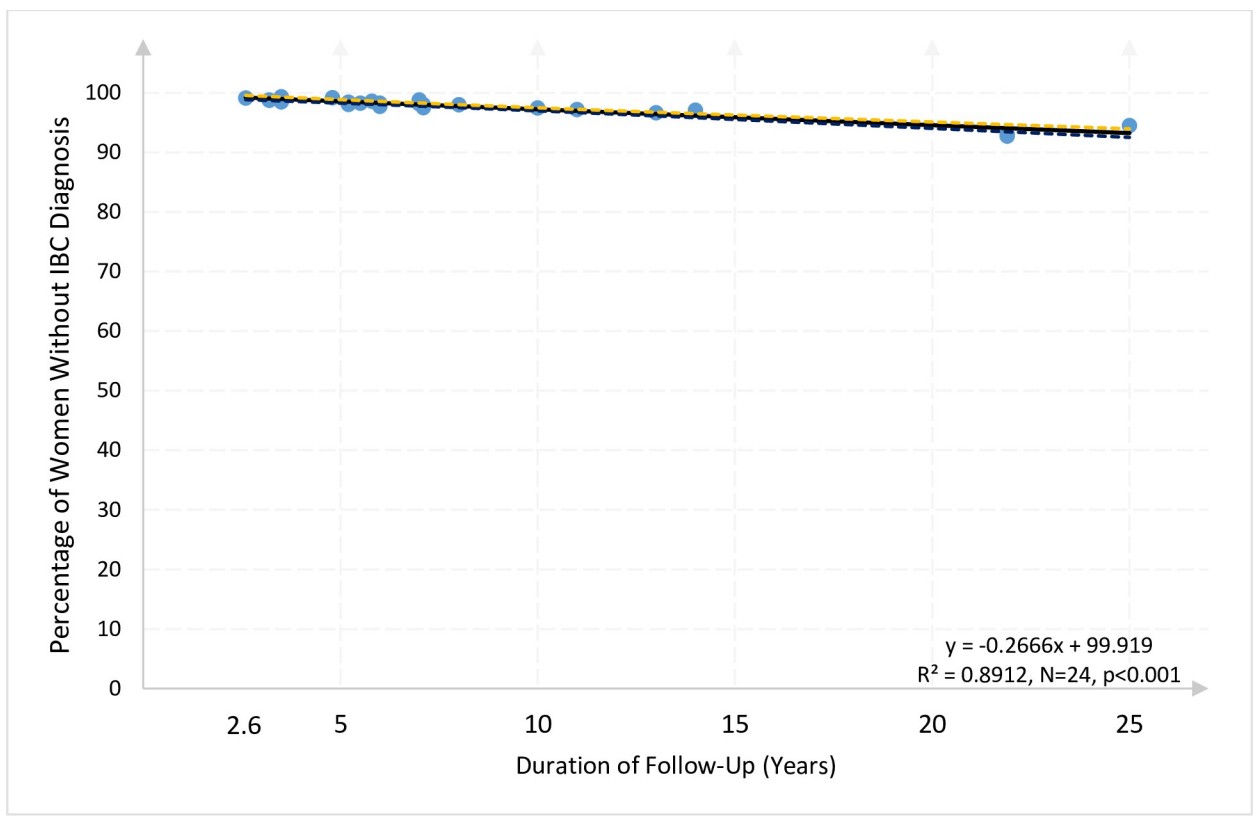

**Fig 1. Percentage of women still free from IBC diagnosis relative to the duration of clinical study (years of follow-up) in 21 studies (24 outcomes) comprising 2,402,672 women.** Scatterplot and regression (solid line) with 95% CI (dotted lines) for the relationship between the probability of remaining free from IBC diagnosis and the follow-up duration based on all 21 studies of pre/postmenopausal women. Each point represents the percentage of women still free from IBC diagnosis at the end of the follow-up period in each study. Each dot describes one observation, *i.e.*, one outcome, from one study. The dashed lines show the 95% confidence interval bounds for the solid regression line.

### Sensitivity analyses

We performed several sensitivity analyses, summarized in Table 4 and two supplements to Table 4. See Online Supplements: S4 and S5 Tables. The reason for such an analysis is to consider whether variability in the sample sizes of different studies, or different subgroups, *e.g.*, the screening or the postmenopausal women, yield different annual rate of change from absence of diagnosis of IBC. Given the heterogeneity of sample sizes across different studies, the error terms in our regression models have unequal variances. Therefore, statistical models that correct for this problem are necessary. The statistical term for this unequal variance problem is "heteroskedasticity".

Table 4 shows the regression results using the generalized least squares (GLS) regression method, which corrects for heteroskedasticity by weighting each of the studies in this systematic review and meta-analysis by its corresponding sample size. Column 1 shows the results using the full sample of 24 outcomes corresponding to the 21 studies comprising 2,402,672 women. Columns 2 to 5 are additional regression models isolating important sub-groups of studies: those that focused on the subsample of women who were at least 50 years old or surgically menopausal at enrollment (Column 2, including only studies 1, 3, 4, 5, 6, 7a, 7b, 8a, 8b, 11, 12, 13, 14, 17, 20, and 21), and those that provided breast cancer screening at enrollment (Column 4, including only studies 1, 4, 5, 7a, 7b, 8a, 8b, 9, 10, 11, 14, 15, 16, 17, 18, 19a and 19b). For completeness, we also provide results for the studies that were not limited to menopausal women (Column 3) and not limited to screened women (Column 5).

**Table 4. Sensitivity analysis.**

|  | (1) | (2) | (3) | (4) | (5) |
|---|---|---|---|---|---|
|  | **Full sample of all studies** | **Subsample of studies limited to post-menopausal women** | **Subsample of studies not limited to menopausal women** | **Subsample of studies limited to screened women** | **Subsample of studies not limited to screened women** |
| Effect of each | **-0.255**\*\*\* | -0.228\*\*\* | -0.254\*\*\* | -0.256\*\*\* | -0.214\*\* |
| additional year of follow-up | **(0.0000)** | (0.0000) | (0.0005) | (0.0000) | (0.0026) |
| Constant | 99.78\*\*\* | 99.71\*\*\* | 99.61\*\*\* | 99.79\*\*\* | 99.52\*\*\* |
|  | (0.0000) | (0.0000) | (0.0000) | (0.0000) | (0.0000) |
| N (Outcomes) | 24 | 16 | 8 | 17 | 7 |
| N (Studies) | 21 | 14 | 7 | 14 | 7 |
| N (Women) | 2,402,672 | 1,808,022 | 594,650 | 2,264,591 | 138,081 |
| R-squared | 0.920 | 0.984 | 0.882 | 0.921 | 0.861 |

Dependent Variable: Percentage of women free of invasive breast cancer (IBC), the "survival rate", at the end of each study.

Statistical $p$-values are shown in parentheses (\*\* $p < 0.01$, \*\*\* $p < 0.001$). The generalized least squares (GLS) regression method was used to correct for heteroskedasticity stemming from unequal error variances across the disparate studies.

Column 1 shows that each additional year of follow-up is associated with an average decline in the "survival rate" of about one-fourth of one percent (0.255 percentage points) per year. Expressed in non-technical language; about ¼ of 1% of women per year lose their freedom from a diagnosis of IBC or 99 ¾% retain their freedom each year.

Columns 2 and 3 show similar results for the subsamples of studies including only postmenopausal women (N = 16 outcomes, 14 studies, 1,808,022 women) and studies not limited to menopausal women (N = 8 outcomes,7 studies, 594,650 women), respectively. Columns 4 and 5 show similar results for the subsamples of studies including only screened women (N = 17 outcomes, 14 studies, 2,264,591 women) and studies not limited to screened women (N = 7 outcomes, 7 studies, 138,081 women).

The flattest line, *i.e.*, the least decline per year, corresponds to the subsample of studies that included non-screened women (column 5). The coefficient of -0.214 in column 5 indicates that one additional year of follow-up for non-necessarily screened women is associated with an average decline in the "survival rate" of 0.214 percentage points per year, *i.e.*, about one-fifth of one percent decline per year.

The GLS regression line indicates a remarkable flat slope of -0.255 (p < .0001, 95% CI: -0.288,-0.221) for the average yearly decline in the absence of the first IBC diagnosis rate. See column 1 of Table 4. This result reinforces our previous estimate using OLS linear regression uncorrected for heteroskedasticity of about one-fourth of one percent. But this slope now is even slightly flatter (a decline in absence of IBC of 0.255 percent per year instead of a decline of 0.267 percent per year). Subsequently, after four years and accounting for heteroskedasticity, the rate of freedom from the first IBC diagnosis is expected to decline by 1.02% (95% CI: -1.15, -0.88). After 25 years of follow-up, the expected rate of absence of ever being diagnosed with IBC remains relatively high, at 93.41 percent (95% CI: 92.75, 94.07).

Table 5 summarizes the estimated probabilities and confidence intervals of remaining free from an IBC diagnosis for intermediate follow-up periods ranging from 0 to 25 years.

## Postmenopausal women

The subset of 14 studies, which included only postmenopausal women, *i.e.*, those who were either at least 50 years or who were surgically menopausal at enrollment, provided 16 outcomes representing 1,808,022 women. The 14 studies are: #1, #3, #4, #5, #6, #7, #8, #11, #12,

**Table 5. Absence of IBC diagnosis probabilities and C.I. for the full sample of 21 studies.**

| Follow-Up Years | Absence of IBC Diagnosis* | P-Value | 95% Confidence Interval | |
|:---:|:---:|:---:|:---:|:---:|
| 0 | 99.78% | 0.000 | 99.52% | 100.00% |
| 5 | 98.51% | 0.000 | 98.34% | 98.68% |
| 10 | 97.23% | 0.000 | 97.01% | 97.45% |
| 15 | 95.96% | 0.000 | 95.61% | 96.31% |
| 20 | 94.69% | 0.000 | 94.18% | 95.19% |
| 25 | 93.41% | 0.000 | 92.75% | 94.07% |

*  **Absence of IBC Diagnosis** refers to the estimated probability of remaining free from an IBC Diagnosis after the specified number of follow-up years. In non-technical language: What % of these women continue to retain their freedom from ever being diagnosed with IBC. These probabilities correspond to the full sample of 21 studies representing 24 outcomes and 2,402,672 women.

#13, #14, #17, #20, and #21. The estimated regression line yields a predicted slope of -0.228 (p<0.001, 95% CI:-0.245,-0.212). See column 2 of Table 4. The slope indicates that the percentage of postmenopausal women expected to remain with no IBC diagnosis decreases by 0.228 per year, *i.e.*, less than one-fourth of one percentage point per year. Thus 94.002% of the postmenopausal women in these 14 studies would be expected to remain free of an invasive breast cancer diagnosis after 25 years of follow-up (95% CI: 93.661, 94.343). Again, after correcting for heteroskedasticity, the estimated regression line is remarkably flat and slightly flatter and more precise than without accounting for heteroskedasticity (See Fig 2, and see also supplement S4, panel B, column 2, for details).

## Screened women

The subset of 14 studies which included breast screening provided 17 outcomes representing 2,264,591 women: The 14 studies are: #1, #4, #5, #7, #8, #9, #10, #11, #14, #15, #16, #17, #18, and #19. The estimated regression line for the subsample of screened women indicates that for each additional year of study follow-up, the percentage of screened women expected to remain free from an IBC diagnosis decreased by 0.256 (p < .0001, 95% CI: -0.297, -0.214). This model predicts that 93.400% of these pre and postmenopausal screened women would remain IBC diagnosis-free after 25 years of follow-up (95% CI: 92.583, 94.218). See column 4 of Table 4. Again, after correcting for heteroskedasticity, the estimated regression line is remarkably shallow and slightly shallower and more precisely estimated than without accounting for heteroskedasticity (See Fig 3, and see also supplement 1, panel B, column 4, for details).

It is worth noting that the subset of studies not limited to screened women (*i.e.*, studies that allowed non-screened women) exhibited an even flatter regression line, *i.e.*, a smaller decline, per year, in the "survival rate" or the rate at which women remain free from IBC. For this subsample of non-necessarily screened women, the estimated coefficient of -0.214 for the slope indicates that each additional year of follow-up is associated with an average decline in the "survival rate" of almost one-fifth of one percent (0.214 percent) per year. See column 5 of Table 4.

Despite the reduction in sample size in the analysis of the 4 subsamples shown, we obtain remarkably consistent results across different groups of studies, as can be seen with the naked eye by the similarity in the slopes of Figs 1–3. To test more rigorously that indeed there are no significant differences in the slopes across subsamples we conducted a binary-variable statistical analysis, the results of which are offered in Supplement 1 to Table 4. See Online Supplements: S4 Table. This analysis indicates that there are no statistically significant differences in

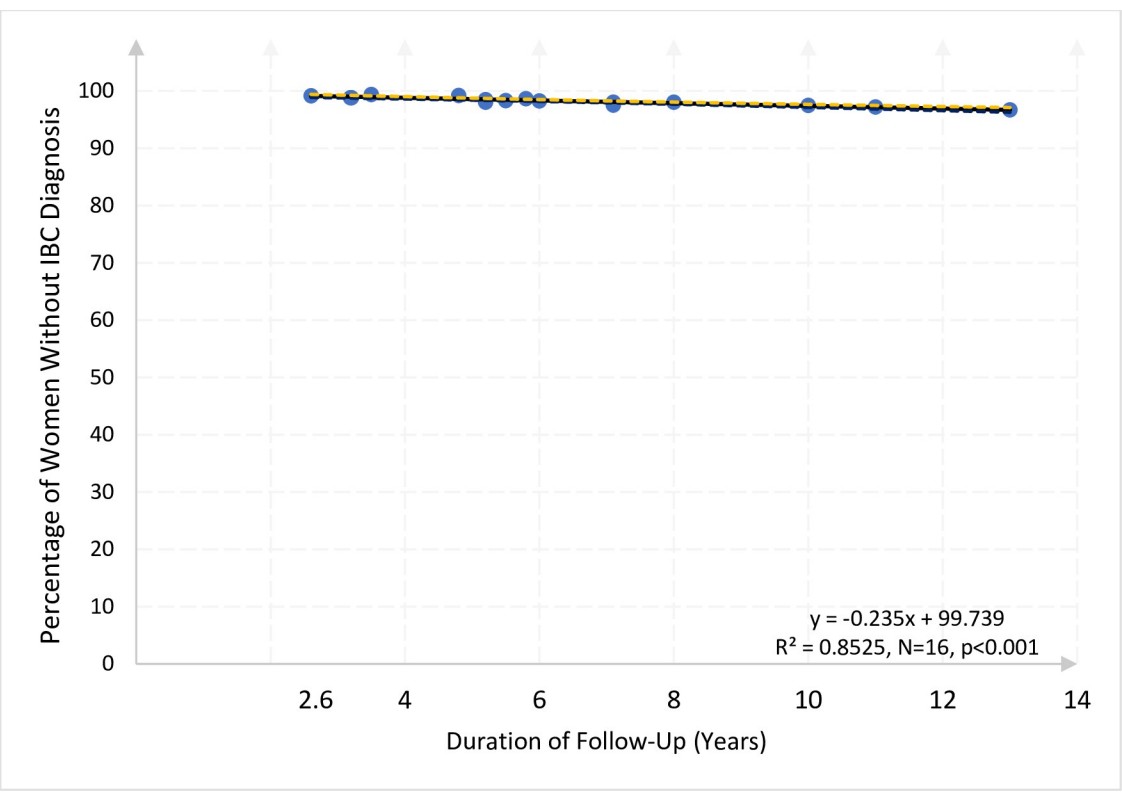

**Fig 2. Percentage of post-menopausal women still free from IBC diagnosis relative to the duration of clinical study (years of follow-up) in 14 studies (16 outcomes) comprising 1,808,022 women.** This figure identifies the 14 studies (16 data points) that included only women who were at least 50 years old or surgically menopausal at enrollment. Studies 1, 3, 4, 5, 6, 7 (a and b), 8 (a and b), 11, 12, 13, 14, 17, 20, and 21. Each point represents the percentage of women still free from IBC diagnosis at the end of the follow-up period in each study. Each dot describes one observation, *i.e.*, one outcome, from one study. The dashed lines show the 95% confidence interval bounds for the solid regression line.

the slopes across subsamples. This result essentially confirms that the flat slopes of Figs 1–3 are not statistically different from each other and not significantly different than the slope coefficients obtained in Table 4 (columns 1 to 5). In sum, these coefficients indicate a decline in the "survival rate" between one-fourth and one-fifth of one percent per year of follow-up.

## Potential effects of outliers and the UK Million Women Study

Our results not only were consistent across subsamples but also robust to the exclusion of potentially influential outcomes. One such important potential concern could be the over-representativeness of the UK Million Women Study. With 1,084,110 women included in this single study, it represents 45.12% of our total sample of women. In supplement 2 to Table 4 (see Online Supplements: S5 Table) we present parallel results to those of Table 4 but excluding the UK Million Women Study. Using the new "full sample" of 20 studies (23 outcomes representing 1,318,562 women), the new model has an estimated slope coefficient of -0.252 (virtually unchanged from the model that included the UK Million Women Study which yielded a slope of -0.255). That means that each additional year of follow-up for women in studies other than the UK Million Women Study is associated with an average decline in the "survival rate" of nearly one-fourth of one percent (0.252 percentage points) per year. See column 1.

In models estimated on important subgroups of women that additionally excluded the UK Million Women Study, the flattest lines, *i.e.*, the least decline per year, correspond to the

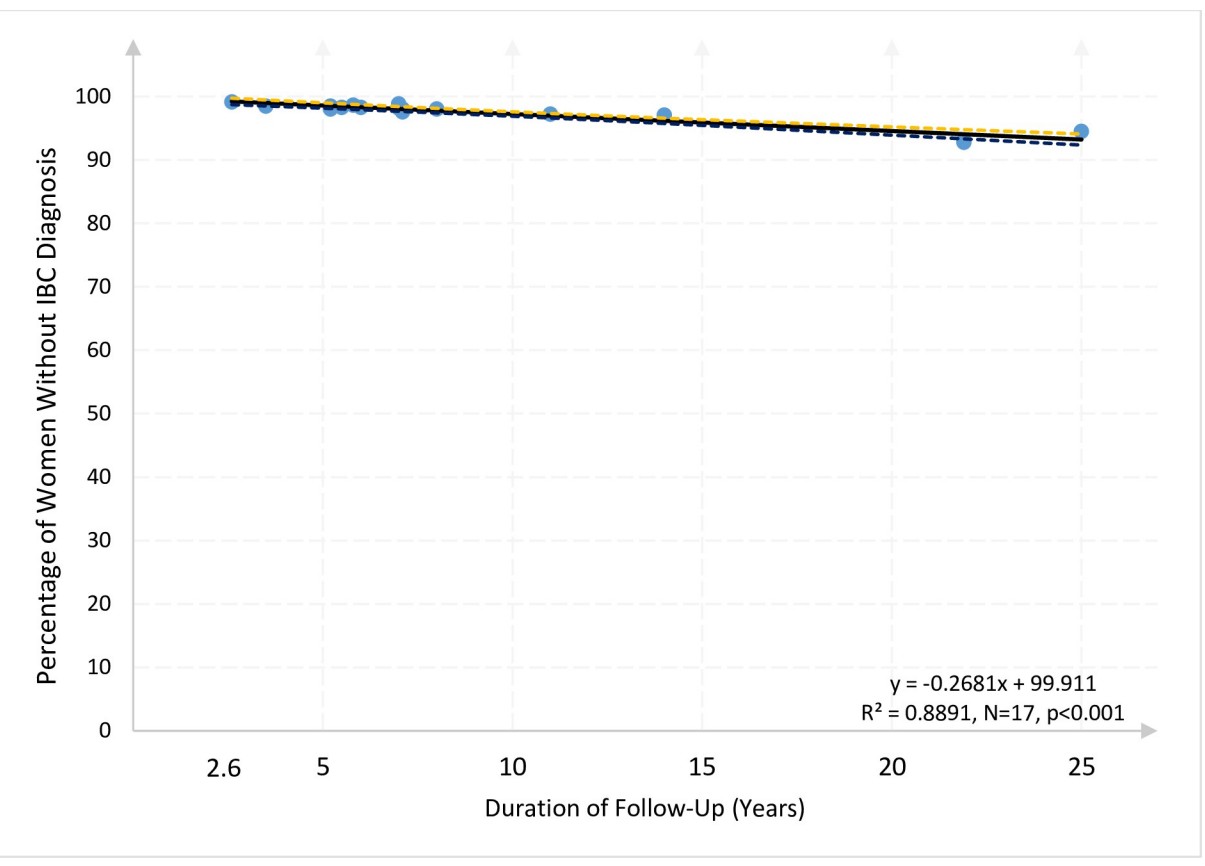

**Fig 3. Percentage of screened women still free from IBC diagnosis relative to the duration of clinical study (years of follow-up) in 14 studies (17 outcomes) comprising 2,264,591 women.** This figure identifies the 14 studies (17 data points) that included only women who were screened at enrollment. Studies 1, 4, 5, 7 (a and b), 8 (a and b), 9, 10, 11, 14, 15, 16, 17, 18, and 19 (a and b). Each point represents the percentage of women still free from IBC diagnosis at the end of the follow-up period in each study. Each dot describes one observation, *i.e.*, one outcome, from one study. The dashed lines show the 95% confidence interval bounds for the solid regression line.

subsamples of post-menopausal women (column 2) and non-necessarily screened women (column 5). The coefficient of -0.209 in column 2 indicates that each additional year of follow-up for post-menopausal women not in the UK Million Women Study is associated with an average decline in the "survival rate" of 0.209 percentage points per year, *i.e.*, about one-fifth of one percent decline per year. These results indicate that the main findings of this research are not driven by the over-representation of women in the UK Million Women Study. These statistical tests corroborate the visual pattern of Figs 1–3, in which all outcomes are very close to the fitted OLS regression line and there are no curving patterns or outliers driving the results.

## Potential bias due to age at enrollment, sample size, and study duration

Additional potential biases were tested considering the impact of differences in sample sizes or differences in the minimum calendar, *i.e.*, chronological, age of participating women. We asked whether smaller or larger sample sizes distort the results and whether studies that enrolled only women at least 50 years of age yielded different results than the ones that included younger pre and postmenopausal women.

Tests for such bias showed either minimal or no differences. A multiple regression model using the full sample of 24 outcomes with the independent variables being the number of

follow-up years and number of women enrolled in each study yielded virtually the same annual decline of -0.253 (95% CI: -0.296, 0.209; p < .000) in the percentage of women remaining free of an IBC diagnosis when holding the number of women enrolled constant. In turn, the impact of the number of women enrolled on the percentage of women expected to remain free from IBC diagnosis, holding the number of follow-up years constant, was statistically insignificant and negligible in magnitude: 0.0% (95% CI: 0.00, 0.00; p<0.80).

Similar results occurred for the multiple regression model incorporating the number of follow-up years and the minimum chronological age of the women at enrollment. There was no significant change in the estimated annual decline in percentage of IBC-free women holding the minimum chronological age constant. The estimated slope coefficient was -0.246 (95% CI: -0.280, -0.211; p < .0001). In turn, the impact on IBC diagnoses stemming from the minimum chronological age at enrollment, holding the number of follow-up years constant, was not statistically different from zero (95% CI: -0.009, 0.070, p>0.1). Our results indicate that no observable factor that could have altered the annual predicted rate of decline in IBC-free women did so.

## Discussion

Invasive breast cancer was uncommon in 21 rigorous studies that set out to search for it: 94 percent of postmenopausal women with no IBC prior history are estimated to remain IBC-diagnosis-free over their next 25 years; *i.e.*, the end of life for many. When pre and postmenopausal women are included together, the regression statistic is just about half a percent lower: 93.41% are expected to remain IBC free over their next 25 years.

Despite the geographic and cultural heterogeneity of these samples, we saw a clear pattern in the data of 2,402,672 women from 21 diverse peer-reviewed studies. Plotted over a 25-year course, remaining IBC-free showed a persistent linear relationship for the entire group of women. Only about ¼ of 1% (0.255%) would be expected to be newly diagnosed per year after their entry and initial screening.

But even this finding may be too pessimistic because over-diagnosing did occur (e.g., Table 2, Study 16). "Overdiagnosis," the finding of a real disease which would never have caused any problem, is now recognized as occurring in as high as 50% of these early detections [25, 29, 30]. Overdiagnosis leads to overtreatment that diminishes the quality of life without adding real benefit. A biopsy is psychologically damaging, even with a benign result [31].

Furthermore, the prospects for healthy women may be even better. Women in these 21 studies were not preselected for good health habits. Some studies included women with compromised health conditions, like the WHI (Table 2, studies 7 and 8) with its obese and overweight populations on statins. Therefore, the estimated probabilities of remaining free from an IBC diagnosis up to 25 years (see Table 5) may be interpreted as a lower boundary for the corresponding true probabilities.

In contrast to the ~6% 25-year-risk estimate here, prevailing calculations for the general population project much higher risk estimates. According to 2019 online NIH reporting, "the lifetime risk estimates for the general population, suggests 12.3 percent of women will develop breast cancer during their lives" [32]. Similarly, in 2019, the American Cancer Society posted this statement: "As reported in previous years, a female's probability of developing invasive breast cancer in her lifetime continues to be 12.4 percent or 1 in 8 women" [33]. While lifetime estimates precede the premenopausal period we examined, women older than 39 accounted for 94.7% of all invasive breast cancer cases according to CDC data [34].

We have previously detailed how these lifetime predictions do not withstand scrutiny because of the way the data are collected and then interpreted [1]. First, the CDC considers each primary tumor and recurrence with a different histology to be a new case, which can

yield multiple case counts in the same woman. Counting cases this way inflates the numerator of the fraction that is used to calculate the incidence rate. Second, the denominator of the fraction is deflated because it is based on census estimates which are skewed toward undercounting.

For example, in Marin County, California, the 2000 census-estimators made an error, underestimating the number of household members by 32%. This increased the calculated breast cancer rate among women 50–69 years by an astonishing 50% [35]. Although these undercounts and census-related estimation problems have been acknowledged, they continue [4, 36–39]. Our results are very encouraging and suggest a much larger population of healthy postmenopausal women than is generally recognized.

Future corroboration of these findings using registry data, when available, would be valuable. Data from those registries that can specify the number of first IBC cases and the population size should enable calculation of cumulative total incidence or absence of IBC over a given span of years of age, by totaling the incidence in each age group. The expectation is that such alternative analysis would yield outcomes similar to those obtained in the current analysis of these 21 studies of 24 outcomes.

The recent changes in the current medical context have resulted in an increased need to accurately inform patients and providers about the true value of screening. The results of this paper represent an important milestone in that direction.

## Limitations

The results of the prospective method we used apply only to those women who have no prior history of IBC diagnosis and meet all five selection criteria, including pre/postmenopausal status. We did not address risk estimates for women at high risk with known genetic markers. Nor can we evaluate groups of women not covered by these 21 studies, such as poor women in underserved populations. The present study also cannot specify how much greater freedom from a diagnosis is experienced by women with optimal health habits. A number of the 21 studies included women who were overweight or obese and none selected for women who practiced regular exercise and avoided excessive alcohol consumption.

The Clinical Practice Guidelines, used by different providers in 2019, are inconsistent with each other and vary depending on the source [40]. But they would be overprescribing mammogram frequency for menopausal women if based on the Retrospective Method of risk estimating used by the CDC and American Cancer Society.

This meta-analysis and systematic review did not exclude studies based on the prior diagnosis of ductal carcinoma in situ (DCIS). DCIS has been shown to elevate the subsequent findings of IBC [41]. However, if DCIS findings did elevate the subsequent findings of a 1st IBC, then such a bias would elevate the 1st IBC incidence found. For screened women, this would bias our results towards a larger incidence of IBC (or towards lower absence of IBC). But the incidence we are finding is already low (the "IBC absence" we are finding is very high). This is despite, not because of, this potential bias. Therefore, in this regard, our results provide a lower boundary for the "survival rate". This limitation then is that absence of an IBC diagnosis would actually be a little bit higher than we could identify in the mammogram screened samples.

We recognize that each of the 21 studies was developed for a different purpose, but the stability of the outcome appears to be a strength of our investigation rather than a limitation. These 21 widely diverse studies allow us to form a striking picture, as shown in Figs 1–3 and statistic outcome Tables 4 and 5, of how women with no prior IBC actually fared as they were followed up.

The 21 studies did not systematically test their cohorts for IBC outcome as it related to some of the known risk factors for invasive breast cancer: high alcohol consumption, obesity, certain HRT regimens that include synthetic progestins rather than those regimens that use progesterone, inadequate daily exercise habits, deficient Vitamin D3, and cumulative total radiation doses to the breasts [42]. While these are each recognized, it remains for future researchers to examine whether behavioral change can increase the freedom from IBC diagnosis. Case control trials would be helpful but difficult to manage.

Data on family history of breast cancer were not provided in any consistent way and, to the degree that women were more likely to be enrolled in a screening program because they are currently cancer free and concerned about their own risk, our findings would be biased as generating a higher incidence of IBC than would occur in women without a family history. Likewise, history of a genetic predisposition would also yield the same limitation.

Chronological age calculates the age in years from the date of birth. Biological age reflects the actual physiologic state: whether one is aging faster or slower than the chronologic age would imply. Data on the biological age of the participants, especially those rendered surgically menopausal, were unobservable and therefore unavailable to us.

Castration (ovariectomy) ages a 35 to 45-year-old woman into immediate menopause, making her biologically older than her intact peers. And even hysterectomy, with preservation of the ovaries, accelerates aging. For example, it triggers a demonstrable acceleration of kyphosis and related loss of bone [43–45].

A rigorous study of the effects of biological age would be desirable but it is beyond the scope of this research. Nevertheless, the stability of our results across different subsamples of women suggests that variations in biological age may have, if any, a minimal effect on our findings. Also, using calendar, *i.e.*, chronological, age as a proxy for biological age we observed an insignificant and negligible effect of age on the annual decline in the percentage of women who remained free from a first IBC diagnosis. This suggests that even though this limitation is worth exploring in future research, it is unlikely that the main findings will be altered.

## Conclusions

We found that postmenopausal women with no prior history of IBC diagnosis have a ~94% chance of remaining IBC free over their next 25 years, and believe that, when guiding pre/postmenopausal patients, our numbers should replace the much higher estimates of breast cancer risk still in common use.

Risk estimates based on registries from cancer cases that exclude women without IBC do not appear to apply to the general population of menopausal women. Screening guidelines might be amended for women at low risk of breast cancer to avoid the dangers of overdiagnosis.

### Potential clinical value

The ultimate goal of any screening test is to save lives. More clinicians are recognizing a lack of mortality benefit from mammogram screening in the normal population after age 40 [29, 30, 46]. Our data analysis demonstrates that the prevalence of breast cancer is not as great as the usual "1 in 8" mantra suggests. Our study should give comfort to those women and providers who choose to do fewer mammogram screenings. Health care leaders and women should reassess the value and frequency of any screening test as we learn more about its strengths and flaws over time [47]. These more precise estimates of a pre/postmenopausal woman's likelihood of remaining free of invasive breast cancer should permit providers to offer more tailored guidance to these patients.

## Supporting information

**S1 Table. The PRISMA abstracts checklist.**
(XLSX)

**S2 Table. Steps 1, 2 and 3 of the first records search.**
(DOCX)

**S3 Table. The steps of the 2012–2019 records search.**
(DOCX)

**S4 Table. S1–S4 Tables.** Extended sensitivity analysis using binary variables.
(DOCX)

**S5 Table. S2–S4 Tables.** Regression analysis excluding the UK Million Women Study.
(DOCX)

**S1 Text. Detailed methods of successive literature searches.**
(DOCX)

## Acknowledgments

The authors express their appreciation to Dr. Brent Mattingly, Chair of Psychology at Ursinus College for his statistical support and the student research interns Ryan Walker and Molly Divis for ongoing manuscript preparation and former student intern, now consultant, Paige Szmodis whose ongoing research assistance helped ferret out the updated search process and papers to review, and finally to Thomas Quay, Esq. for his iterative readings to help assure clarity in general English usage.

## Author Contributions

**Conceptualization:** Winnifred Cutler, James Kolter, Catherine Chambliss, Heather O'Neill, Hugo M. Montesinos-Yufa.

**Data curation:** Winnifred Cutler, James Kolter.

**Formal analysis:** Heather O'Neill, Hugo M. Montesinos-Yufa.

**Funding acquisition:** Winnifred Cutler.

**Investigation:** Winnifred Cutler, James Kolter, Catherine Chambliss, Heather O'Neill, Hugo M. Montesinos-Yufa.

**Methodology:** Heather O'Neill, Hugo M. Montesinos-Yufa.

**Project administration:** Winnifred Cutler, Catherine Chambliss.

**Resources:** Winnifred Cutler, Catherine Chambliss.

**Supervision:** Winnifred Cutler.

**Validation:** Hugo M. Montesinos-Yufa.

**Visualization:** Hugo M. Montesinos-Yufa.

**Writing – original draft:** Winnifred Cutler, James Kolter, Catherine Chambliss, Heather O'Neill, Hugo M. Montesinos-Yufa.

**Writing – review & editing:** Winnifred Cutler, James Kolter, Catherine Chambliss, Heather O'Neill, Hugo M. Montesinos-Yufa.

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
