## [Decision Letter · Decision Letter 0]

7 May 2020

PONE-D-20-03266

Long term freedom from invasive breast cancer diagnosis in 2,402,672 peri and postmenopausal women: a systematic review and meta-analysis

PLOS ONE

Dear Dr Cutler,

Thank you for submitting your manuscript to PLOS ONE. After careful consideration, we feel that it has merit but does not fully meet PLOS ONE’s publication criteria as it currently stands. Therefore, we invite you to submit a revised version of the manuscript that addresses the points raised during the review process.

We would appreciate receiving your revised manuscript by June 8th, 2020. To enhance the reproducibility of your results, we recommend that if applicable you deposit your laboratory protocols in protocols.io, where a protocol can be assigned its own identifier (DOI) such that it can be cited independently in the future. For instructions see: http://journals.plos.org/plosone/s/submission-guidelines#loc-laboratory-protocols

We look forward to receiving your revised manuscript.

Kind regards,

Magdalena Grce, PhD

Academic Editor

PLOS ONE

Journal Requirements:

'Funding: The author(s) received no specific funding for this work. Athena Institute provided funding in the form of salary for author Winnifred Cutler and the salaries of the student research intern Ryan Walker for ongoing manuscript preparation and former student intern, now consultant, Paige Szmodis for her literature search work.  Each coauthor provided, without compensation, the time and energy both to the analysis and the writing. Athena Institute had no further role in study design, data collection and analysis, decision to publish, or preparation of the manuscript.'

We note that one or more of the authors have an affiliation to the commercial funders of this research study: Athena Institute Inc

4. Please remove your figure from within your manuscript file, leaving only the individual TIFF/EPS image file, uploaded separately.  This will be automatically included in the reviewers’ PDF.

Reviewers' comments:

Reviewer's Responses to Questions

**Comments to the Author**

1. Is the manuscript technically sound, and do the data support the conclusions?

Reviewer #1: Yes

2. Has the statistical analysis been performed appropriately and rigorously? 

Reviewer #1: Yes

3. Have the authors made all data underlying the findings in their manuscript fully available?

Reviewer #1: Yes

4. Is the manuscript presented in an intelligible fashion and written in standard English?

Reviewer #1: Yes

5. Review Comments to the Author

Reviewer #1: This manuscript that provides an alternative to breast cancer risk using cohort studies across the globe makes an intriguing assertion. Specifically, the calculation used to assess breast cancer risk, and therefore recommendations for mammogram frequency though cancer registry data is flawed for menopausal women. Overall, this is a great paper but needs some edits/analyses to further strengthen this. It does provide a strong and important message.

Some specific comments are below

Line 81. I don’t understand how many of the participants enrolled at younger ages (age 39, 35, 40, etc) are considered perimenopause without there being clear indication in the manuscripts that women are experiencing hormonal changes. Given the average age is early 50s and symptoms associated with hormonal changes averages 6-8 years, many of the younger women are likely not in the perimenopausal phase. Unless this was determined, then the inclusion is just women with no IBC hx at enrollment would be more accurate. Recommend not using the perimenopausal to partially describe this population.

Line 91. Beginning with the statement “unfortunately…” This is unclear what the authors are saying. Please clarify.

The most valid statement is the point that cancer registry calculator using new incidence case not new person with IBC as numerator; I was not aware that was how the numerator was calculate. However if this was changed to be first IBC what would the risk be? To me adding this piece of information would significantly strengthen the paper.

Table 1 is a nice outline of the differences. My understanding was the cancer registry used the number of cases, not an estimate of number of cases. Do you mean the current year’s estimate? They always give an estimate for the current year’s since the case ascertainment has a long lag tail for inclusion. Point 4 is unclear to me. If the registry uses incident IBC, why is followup needed. In xx year, this is the number of women with IBC as numerator and denominator in as estimate of number of women who have to possibility of being dx with IBC.

Table 2 is a nice outline. However, I would delete the repetitive statement in bold, ‘1st incidence of IBC’ for each study. There was some inconsistencies in description; some used menopausal status, some used study type (screened or not screened), some used HRT, some used menopausal symptom medication. The purpose of this table is unclear since similar items were not included for each study---it lacks consistency. For example, many start with bolded study type (Cohort or RCT, but then others drift such as 11, record review (isn’t that a retrospective cohort study) or #15 was ?? or #20 and #21 did not specify study design. Some use menopausal status and other used age (with some clearly menopausal). No indication of any of these interventions that could protect against IBC. Please address/add info for consistency.

For the RCTs was only controls used in the analyses? It seems not, and therefore it makes me wonder if the intervention affected IBC. Any way to address this concern?

The other concern about these cohort studies in which screening was a key aspect AND the outcome for this study was IBC is that we all know screening can pick up in situ BC so this seems to influence the numerator. How was this bias handled?

Table 2 provides important information. Can you do the age adjusted rate here instead of just crude percent since age is one of the most important variables in cancer dx.

Fig 1, just include 90% to 100% on the y-axis so this can be read for easily. The title of y says percent of women without IBC dx and it is at 100% . It doesn’t make sense that all women or 50% or 70% of women would have IBC at baseline. It looks like it starts at baseline with 0% free IBC and goes to around 7-8% with IBC at 25 years. The description is better than the image and the image can be improved upon.

From what I could understand from Table 4, everything is significant at <0.05. That is not surprising given the sample size. Given this sample size, it seems reasonable to only consider significance at the <0.001. Overall this table is unreadable. Please use footnotes to help describe all our models. Please add number of participants in each model since it seems this varied. Years means ? PM=1 means; PM=0 means? Overall this is a very unclear table.

I’m struggling with the concept of free (probably not use freedom) from IBC dx, that does not take into consideration age of participant. Another piece of the data that makes me worried is the over-representativeness of the UK Million Women Study. Almost ½ of the participants come from this group and the fu was the shortest at 2.6 years. If the analyses were run without this study, would similar results be reported? I read further that you did an analysis where age and sample size was addressed (lines 217-235). I don’t fully understand the test. How was the independent variables constructed? Number of women enrolled entered into the model as ? Put years studied (means followup period?) and sample size … and found similar result, ‘holding number of women constant’. I just don’t follow the entire paragraph. As far as I can tell this does not address age + followup period as important in the analysis.

Probably would not use the word ‘bad health habits’ since this is a judgmental word. Line 250

The strongest argument for this paper is that the conventional use of a numerator for determining bc risk may over-inflate the risk since recurrence is well known. Without knowing what the rate would be by changing the numerator using cancer registry data is a significant limitation in the argument made in this paper. I also believe the ACS considers both IBC and in situ. About 12-15% of bc are insitu with strong mammogram uptake and this should be noted as part of their calculation. This also reflects a limitation in this analysis, women who were dx with in situ through use of mammograms, never developed IBC and were not ‘counted’ in the statistics. It seems a bit like comparing apples and oranges.

Please provide some reference to support the statement on line 268 that “census estimates are skewed towards undercounting.” Using one example in which an error was made and acknowledged as an error is not sufficient to make this global statement. More documentation of this is needed.

The limitation section needs to include a few of the comments made in this review. Many of the known risk factors may or may not have been included in the studies, such as family history of bc. Adding information about the bc risk factors (prevalences) by study would be a great addition and likely readily available. This could strengthen the analysis since it represent the population or does not. I would suspect it doesn’t represent lower educational attainment, or minorities (for US). If the study descriptions are not truly representative this should be included in the limitations and if it does, then that is a strength that should be noted.

Overall, I think this is a great analysis and just needs to be tighten up to provide a very strong message.

6. PLOS authors have the option to publish the peer review history of their article (what does this mean?). If published, this will include your full peer review and any attached files.

Reviewer #1: No

---

## [Author Response · Author response to Decision Letter 0]

9 Jun 2020

We have attached a file to download entitled Response to Reviewers

---

## [Decision Letter · Decision Letter 1]

10 Jul 2020

PONE-D-20-03266R1

Long term freedom from invasive breast cancer diagnosis in 2,402,672 peri and postmenopausal women: a systematic review and meta-analysis

PLOS ONE

Dear Dr. Cutler,

Thank you for submitting your manuscript to PLOS ONE. After careful consideration, we feel that it has merit but does not fully meet PLOS ONE’s publication criteria as it currently stands. Therefore, we invite you to submit a revised version of the manuscript that addresses the points raised during the review process.

We look forward to receiving your revised manuscript.

Kind regards,

Magdalena Grce, PhD

Academic Editor

PLOS ONE

Reviewers' comments:

Reviewer's Responses to Questions

**Comments to the Author**

1. If the authors have adequately addressed your comments raised in a previous round of review and you feel that this manuscript is now acceptable for publication, you may indicate that here to bypass the “Comments to the Author” section, enter your conflict of interest statement in the “Confidential to Editor” section, and submit your "Accept" recommendation.

Reviewer #1: (No Response)

2. Is the manuscript technically sound, and do the data support the conclusions?

Reviewer #1: Yes

3. Has the statistical analysis been performed appropriately and rigorously? 

Reviewer #1: Yes

4. Have the authors made all data underlying the findings in their manuscript fully available?

Reviewer #1: Yes

5. Is the manuscript presented in an intelligible fashion and written in standard English?

Reviewer #1: Yes

6. Review Comments to the Author

Reviewer #1: Thanks for the extensive responses.

I appreciate the response to the issue of using peri-menopausal. I am puzzled by the insistence on using this term, including in the title when it does not capture the characteristics of the women and muddles the picture. As mentioned by the reviewers “For the 3 studies that enrolled women who were at least 40:#15,18, and 19 we also looked closely at each before qualifying them as studying women in their peri/postmenopausal era, since the term “perimenopausal” is a loose one – but does convey our intent.” No idea what the authors mean by ‘the intent’. I don’t get it since these 3 studies start at women who are 40 years old. We can argue about this, but ultimately you can choose to use this ‘loose term’ extensively and in the title but be aware that if I read this, I would question your work. My thinking would be, ‘if they ‘misclassify’ women by assigning them a phase of menopausal status that they are not definitely experiencing, then what else were the authors less precise about. At minimum this should be defined in the first paragraph of the paper—perimenopausal is defined in this paper as any women aged 40 and older who has not been classified as menopausal.

In response to the request about calculating rate using only first case, the rationale to not do this is not compelling. Of course at any hospital level, your statement (“And we found that accurate information about prior cases from different hospitals the patient had used was not reported or often even available.”) is not a surprising finding. We are all aware of the fragmentation of health care in the US. Not sure about the other counties.

“Therefore, to your point “if this was changed to be first IBC what would the risk be?” we really cannot

estimate the risk using the retrospective method that the CDC uses because the raw data are not collected,

and they are unavailable.” This is a puzzling statement. What do you mean ‘raw data are not collected.’ It is a federal mandate to report cancer for any patient and in requesting data from the registry, first dx is a field that must be specified. I don’t understand your rationale for gold standard; using a registry that captures all cases is a much better method to estimate risk, if the numerator excludes those who have a recurrence. I thought this was one of your major points. If not, then I am truly confused. Prospective studies, unless very large and representative of the population AND have incredibly high retention rates (not happening in this era of research) are not the gold standard to determine risk. This paragraph puts a spanner in the paper, to me. Why would you not use cancer registry data but have it set up, in the future, to do a ‘better job’ of calculating risk? Or do you think it is impossible to use cancer registry data for estimating risk? What am I missing since this paragraph suggests that we should basically throw out cancer registry data for estimating risk. Is this really your conclusion?

In Table 1, #1: add the denominator is xxx. And for retrospective cancer registry (“estimated number of eligible women in the census for the ….” The key point here is the number of eligible women are estimated. Add ‘eligible’ women to prospective. To me that is the different in #1. One is exact number of eligible women and the other is an estimate of eligible for the denominator.

#2 implies a different way in which women’s bc dx was determined but some/many/most of the studies use the same system. I didn’t read each study, but I would be very surprised if the very large prospective studies did not use cancer registries as linkage to determine dx. Is this incorrect? I just randomly picked one study (Finnish) and they used the registry for followup bc dx.

I want to reiterate that this is a GREAT study and work. I just want to support the representation of the data described to be precise and without question so that it gets the attention it needs vs distracting comments that take away from the message. The only difference between the 2 systems is one uses a list and matches it to the registry and the other just uses the registry list.

#3 This is the key point of the study, to me.

#4 I’m not sure what you mean here. The registries enter date of dx. Longitudinal studies enter date of dx. Longitudinal studies somewhat arbitrarily enroll women on a certain date. Why is that meaningful for this analysis. By default, cancer registries ‘follow’ all women from year of eligibility to date of dx since it is a registry. Cumulative incidence is the number of new events or cases of disease divided by the total number of individuals in the population at risk for a specific time interval. This again gets back to the basic question as to whether the cancer registries can differentiate between first and subsequent cancer dx. I just don’t get this one as a difference between the 2 systems in a meaningful way.

#5 is kind of redundant. The ‘known follow-up period’ isn’t useful info for calculating incidence rate. This is using #1 and #3

Table 4. What is observation? Is this cancer dx? Besides this question the table is so much easier to understand. Thanks

I disagree with the final sentence since the finding of this great analysis was not about the patient’s anxiety and optimism but rather the risk of breast cancer diagnosis by the general population. One could easily argue that this finding can have the opposite effect on the general population of reducing compliance with breast cancer screening since it looks like the risk is much lower than publicly portrayed, from these analyses. If these findings are accepted as norm, we could reasonably go back to the 1980s rates of screening and subsequent more late stage cancer dx. I would not call that scenario a significant societal benefit. I’m not convinced that low risk women (such as no family hx) are anxious about being diagnosed with breast cancer. To me this last sentence distracts from the findings by injecting a personal perspective that could easily be countered.

I also note that this nuance, non-neutral word ‘freedom’ is also used in table footnote. I already indicated my perspective that using this word choice may do a disservice to this work due to inserting non-neutral language.

I would appreciate a definition with maybe an example to understand chronological age vs participant’s age, in the paper.

Appreciate seeing the formulas for the analyses since that was much easier to follow than the description as well as a more information in the limitation section.

7. PLOS authors have the option to publish the peer review history of their article (what does this mean?). If published, this will include your full peer review and any attached files.

Reviewer #1: No

---

## [Author Response · Author response to Decision Letter 1]

23 Jul 2020

I have already added a July 23, 2020 letter to the download list but cut and copy here too for redundancy

Response from Reviewer 7/10/20

6. Review Comments to the Author

Reviewer #1: Thanks for the extensive responses.

1] I appreciate the response to the issue of using peri-menopausal. I am puzzled by the insistence on using this term, including in the title when it does not capture the characteristics of the women and muddles the picture. As mentioned by the reviewers “For the 3 studies that enrolled women who were at least 40:#15,18, and 19 we also looked closely at each before qualifying them as studying women in their peri/postmenopausal era, since the term “perimenopausal” is a loose one – but does convey our intent.” No idea what the authors mean by ‘the intent’. I don’t get it since these 3 studies start at women who are 40 years old. We can argue about this, but ultimately you can choose to use this ‘loose term’ extensively and in the title but be aware that if I read this, I would question your work. My thinking would be, ‘if they ‘misclassify’ women by assigning them a phase of menopausal status that they are not definitely experiencing, then what else were the authors less precise about. At minimum this should be defined in the first paragraph of the paper—perimenopausal is defined in this paper as any women aged 40 and older who has not been classified as menopausal.

Response 1: We agree with you and have changed “peri” to “pre” throughout the entire manuscript. 

We confirmed that standard usage of the 2 terms renders this change essential.

• Premenopausal | Definition of Premenopausal by Merriam ... 

• https://www.merriam-webster.com/dictionary/premenopausalPremenopausalhttps://www.merriam-webster.com/dictionary/premenopausalPremenopausal definition is - of, relating to, or being in the period preceding menopause. ... https://www.merriam-webster.com/dictionary/premenopausal. 

• Perimenopause | Definition of Perimenopause by Merriam ... 

• https://www.merriam-webster.com/dictionary/perimenopausePerimenopause definition is - the period around the onset of menopause that is often marked by various physical signs (such as hot flashes and menstrual ... 

Thank you for reemphasizing this point. 

……………………………

2] In response to the request about calculating rate using only first case, the rationale to not do this is not compelling. Of course, at any hospital level, your statement (“And we found that accurate information about prior cases from different hospitals the patient had used was not reported or often even available.”) is not a surprising finding. We are all aware of the fragmentation of health care in the US. Not sure about the other counties.

“Therefore, to your point “if this was changed to be first IBC what would the risk be?” we really cannot estimate the risk using the retrospective method that the CDC uses because the raw data are not collected, and they are unavailable.” This is a puzzling statement. What do you mean ‘raw data are not collected.’ It is a federal mandate to report cancer for any patient and in requesting data from the registry, first dx is a field that must be specified. I don’t understand your rationale for gold standard; using a registry that captures all cases is a much better method to estimate risk, if the numerator excludes those who have a recurrence. I thought this was one of your major points. If not, then I am truly confused. Prospective studies, unless very large and representative of the population AND have incredibly high retention rates (not happening in this era of research) are not the gold standard to determine risk. This paragraph puts a spanner in the paper, to me. Why would you not use cancer registry data but have it set up, in the future, to do a ‘better job’ of calculating risk? Or do you think it is impossible to use cancer registry data for estimating risk? What am I missing since this paragraph suggests that we should basically throw out cancer registry data for estimating risk. Is this really your conclusion?

Response 2: 

Yes, you are correct that this is one of our major points. And I now SEE what you mean about registry data. We agree that for those registries that can define the number of 1st IBC cases, and the population size in question, and where the resident population is stable it would be doable. Researchers should be able to analyze their results to yield the incidence of 1st cases at each group over 25 years. We believe this would most likely render very similar outcomes to what we have identified in these 21 studies of 24 sets of data, i.e., outcomes. 

It seems that you are referring to health care systems that are more homogenous than those in the US. And you are referring to geographic regions that have none of the worldwide migration we see happening now, with numbers of undocumented persons uncountable all over the world. In such regions, cancer will get recorded if they involve connecting the woman with the health care system. But the number of undocumented persons in the region who do not have a cancer diagnosis, the denominator of the incidence fraction, will be unknown.

In places where the 'universal' health care systems function with rigorously managed registries of all the women in the population such analysis should be possible. Such registry systems are so far ahead of the fragmented health care in US. The CDC only “catches and records cases” but never registers women when they are well to follow up what happens to them. 

Because of variation in data collection methods across countries, we worked to find a prospective method to analyze all studies that enrolled women with no prior IBC, that searched for IBC for a given duration and then reported their rigorous findings. We selected only (and all) those 21 published studies that shared the common features listed in our methods section. We agree with you that some of these 21 studies worked with a usable central registry. But we find others did not. For example, studies 7,8,11, 14 enrolled IBC free women and followed their outcomes independently of registries. 

Now that we understand what you were suggesting, we have inserted this paragraph, preceding the final paragraph in the Discussion Section

Future corroboration of these findings using registry data, when available, would be valuable. Data from those registries that can specify the number of first IBC cases and the population size should enable calculation of cumulative total incidence or absence over a given span of years of age, by totaling the incidence in each age group. The expectation is that such alternative analysis would yield outcomes similar to those obtained in the current analysis of these 21 studies of 24 outcomes.

………………….

3] In Table 1, #1: add the denominator is xxx. And for retrospective cancer registry (“estimated number of eligible women in the census for the ….” The key point here is the number of eligible women are estimated. Add ‘eligible’ women to prospective. To me that is the different in #1. One is exact number of eligible women and the other is an estimate of eligible for the denominator.

Response 3: We have made these changes. 

………………….

4] #2 implies a different way in which women’s bc dx was determined but some/many/most of the studies use the same system. I didn’t read each study, but I would be very surprised if the very large prospective studies did not use cancer registries as linkage to determine dx. Is this incorrect? I just randomly picked one study (Finnish) and they used the registry for followup bc dx.

I want to reiterate that this is a GREAT study and work. I just want to support the representation of the data described to be precise and without question so that it gets the attention it needs vs distracting comments that take away from the message. The only difference between the 2 systems is one uses a list and matches it to the registry and the other just uses the registry list.

Response 4: In the current retrospective method employed by the CDC in the U.S., “participants” (the whole population) are not followed-up in the precise way we are able to achieve using our prospective method. No women are enrolled by the CDC risk estimating method. Only cancer cases are counted into the system. In the USA retrospective methods, the risk calculations do not use individual data directly. In turn, periods of data collection are proxies for the “follow-up” period. Therefore, the cumulative incidence of IBC calculated from the CDC retrospective method can only be seen as an approximation using aggregated data. And even if they could identify only first cases, they would still have the problem of not knowing the population size, i.e., the true denominator of the incidence fraction. 

5] #3 This is the key point of the study, to me.

Response 5: Yes it is one of the key points 

……………..

6] #4 I’m not sure what you mean here. The registries enter date of dx. Longitudinal studies enter date of dx. Longitudinal studies somewhat arbitrarily enroll women on a certain date. Why is that meaningful for this analysis. By default, cancer registries ‘follow’ all women from year of eligibility to date of dx since it is a registry. Cumulative incidence is the number of new events or cases of disease divided by the total number of individuals in the population at risk for a specific time interval. This again gets back to the basic question as to whether the cancer registries can differentiate between first and subsequent cancer dx. I just don’t get this one as a difference between the 2 systems in a meaningful way.

Response 6: see above “Response 2 and Response 4”

…………………..

7]#5 is kind of redundant. The ‘known follow-up period’ isn’t useful info for calculating incidence rate. This is using #1 and #3

Response 7: We prefer, however, to leave it even if it is redundant. It would help the reader put #1 and #3 together to compare side by side the calculation of incidence under each method. 

………………………

8] Table 4. What is observation? Is this cancer dx? Besides this question the table is so much easier to understand. Thanks

Response 8] No. We have changed the word ”observation” throughout the ms to “outcome” and inserted a missing “N” in front of that word in the table. For example, in column 1, the Full Sample, the reader of the table now sees a consistent description. The N equals the Number. For example, in column 1, covering all the studies: There were an ‘N’ of 24 separate outcomes in the 21 studies of 2,402,672 women. The outcomes are the data points that are shown in each figure and in Table 3 : one data point for each study or study subgroup. Thank you for pointing out the confusion since we had been missing the “N’ and we do not want to make the reader guess! So, inserting the missing “N” in front makes it more consistent for the reader of the table to decipher.

…………………………

Table 4 – Sensitivity analysis

Dependent Variable: Percentage of women free of invasive breast cancer (IBC), the “survival” rate, at the end of each study. 

 (1) (2) (3) (4) (5)

 Full 

sample

of all studies Subsample 

of studies limited to post-menopausal women Subsample 

of studies not limited to menopausal women Subsample 

of studies limited to

screened 

women Subsample 

of studies not limited to screened women

Effect of each -0.255*** -0.228*** -0.254*** -0.256*** -0.214**

additional year of follow-up (0.0000) (0.0000) (0.0005) (0.0000) (0.0026)

Constant 99.78*** 99.71*** 99.61*** 99.79*** 99.52***

 (0.0000) (0.0000) (0.0000) (0.0000) (0.0000)

N (Outcomes) 24 16 8 17 7

N (Studies) 21 14 7 14 7

N (Women) 2,402,672 1,808,022 594,650 2,264,591 138,081

R-squared 0.920 0.984 0.882 0.921 0.861

Statistical p-values are shown in parentheses (** p < 0.01, *** p < 0.001). The generalized least squares (GLS) regression method was used to correct for heteroskedasticity stemming from unequal error variances across the disparate studies. 

……………………

9] I disagree with the final sentence since the finding of this great analysis was not about the patient’s anxiety and optimism but rather the risk of breast cancer diagnosis by the general population. One could easily argue that this finding can have the opposite effect on the general population of reducing compliance with breast cancer screening since it looks like the risk is much lower than publicly portrayed, from these analyses. If these findings are accepted as norm, we could reasonably go back to the 1980s rates of screening and subsequent more late stage cancer dx. I would not call that scenario a significant societal benefit. I’m not convinced that low risk women (such as no family hx) are anxious about being diagnosed with breast cancer. To me this last sentence distracts from the findings by injecting a personal perspective that could easily be countered.

I also note that this nuance, non-neutral word ‘freedom’ is also used in table footnote. I already indicated my perspective that using this word choice may do a disservice to this work due to inserting non-neutral language.

Response 9: We agree that the purpose of this paper will be better served by maintaining neutral language as much as possible. We have therefore deleted the last 2 sentences. And we have altered the non-neutral “freedom” term as shown in the title and throughout the ms. We replaced “freedom from” with “absence of” and several other similar variations.

…………………

10] I would appreciate a definition with maybe an example to understand chronological age vs participant’s age, in the paper.

Response 10] 

The chronological age is the participants’ age in this paper. The biologic age of the participants is unknown.

We have altered the paragraph in limitations to add SEVERAL SENTENCES with references. It now says:

Chronological age calculates the age in years from the date of birth. Biological age reflects the actual physiologic state: whether one is aging faster or slower than the chronologic age would imply. Data on the biological age of the participants, especially those rendered surgically menopausal, were unobservable and therefore unavailable to us. 

Castration (ovariectomy) ages a 35 to 45-year-old woman into immediate menopause, making her biologically older than her intact peers. And even hysterectomy, with preservation of the ovaries, accelerates aging. For example, it triggers a demonstrable acceleration of kyphosis and related loss of bone .

………………………

11] Appreciate seeing the formulas for the analyses since that was much easier to follow than the description as well as a more information in the limitation section

Response 11] Thank you for all your comments and suggestions. We think they helped make a better paper and are grateful.

---

## [Editor Report · Decision Letter 2]

6 Aug 2020

Long term absence of invasive breast cancer diagnosis in 2,402,672 pre and postmenopausal women: a systematic review and meta-analysis

PONE-D-20-03266R2

Dear Dr. Cutler,

We’re pleased to inform you that your manuscript has been judged scientifically suitable for publication and will be formally accepted for publication once it meets all outstanding technical requirements.

Kind regards,

Magdalena Grce, PhD

Academic Editor

PLOS ONE
---

## [Editor Report · Acceptance letter]

13 Aug 2020

PONE-D-20-03266R2 

Long term absence of invasive breast cancer diagnosis in 2,402,672 pre and postmenopausal women: *a systematic review and meta-analysis*

Dear Dr. Cutler:

I'm pleased to inform you that your manuscript has been deemed suitable for publication in PLOS ONE. Congratulations! Your manuscript is now with our production department. 

Kind regards, 

on behalf of

Dr. Magdalena Grce 

Academic Editor

PLOS ONE